# Multi-Threaded Sound Propagation Algorithm to Improve Performance on Mobile Devices

**DOI:** 10.3390/s23020973

**Published:** 2023-01-14

**Authors:** Eunjae Kim, Sukwon Choi, Cheong Ghil Kim, Woo-Chan Park

**Affiliations:** 1Department of Computer Science and Engineering, Sejong University, Seoul 05006, Republic of Korea; 2Department of Computer Science, Namseoul University, Cheonan 31020, Republic of Korea

**Keywords:** sound rendering, multi-threaded algorithm, ray tracing

## Abstract

We propose a multi-threaded algorithm that can improve the performance of geometric acoustic (GA)-based sound propagation algorithms in mobile devices. In general, sound propagation algorithms require high computational cost because they perform based on ray tracing algorithms. For this reason, it is difficult to operate sound propagation algorithms in mobile environments. To solve this problem, we processed the early reflection and late reverberation steps in parallel and verified the performance in four scenes based on eight sound sources. The experimental results showed that the performance of the proposed method was on average 1.77 times better than that of the single-threaded method, demonstrating that our algorithm can improve the performance of mobile devices.

## 1. Introduction

Recently, as interest in blockchain/metaverse/XR/VR/MR has increased [1,2], more research to improve the sense of reality and immersion has been conducted. However, many studies have focused only on the visual element. To improve immersion in virtual environments or multimedia applications, high quality auditory as well as visual elements is essential [3], and sound rendering provides users with a higher quality of auditory elements by giving them a better understanding of intuitive spatial cues.

Sound rendering, which produces high-quality audio, consists of two steps: sound propagation and auralization. The former deals with the propagation of sound waves in virtual space, creating impulse responses (IRs) that are encoded with direction, delay, and frequency-dependent attenuation from a source to a listener. The latter convolves prerecorded or synthetically generated dry audio with IRs to generate the final audio signal and output it to an output device such as speakers or headphones.

In general, the sound propagation stage has the highest computational cost and requires the most resources in the entire sound rendering process [4]. There are two main ways to do this. One is a wave-based numerical method and the other is a geometric acoustic (GA) method. The wave-based numerical method numerically calculates wave equations in the time domain [5] or frequency domain [6]. In this way, the computational cost increases exponentially as the scene size and frequencies increase. Although it has the advantage of being able to generate realistic sounds, it is not suitable for real-time applications because it requires considerable time and computing power [7].

In contrast, the GA method uses ray, beam, or frustum tracing to find valid propagation paths, such as direct, transmission, reflection, and diffraction paths, between a listener and a sound source and to estimate multiple reverberation parameters according to space (e.g., size, absorption coefficients). IRs are calculated using the information finally computed through these processes. Therefore, the GA method is suitable for interactive applications because it has a relatively fast processing speed compared to the wave-based numerical method and can track moving source–moving receiver (MS–MR) and geometry scene data at every frame.

Most of the current studies employing the GA method use the high computational power on the PC platform to accelerate the sound propagation algorithm, thereby achieving real-time rates (e.g., 30 fps) [8,9]. However, it is very challenging to perform sound propagation algorithms at real-time rates in mobile devices with computing power and memory constraints [10].

Moreover, numerous studies use excessive CPU (four cores or more) or GPU resources only for the sound propagation algorithm. A sound propagation algorithm that uses many CPU cores makes it difficult to process tasks other than sound propagation, which is impractical. Likewise, a sound propagation algorithm using a GPU is unfeasible in real-time applications such as games because it is difficult to use GPU resources for visual rendering.

For the above reasons, a sound propagation algorithm in a mobile device environment must be processed based on a CPU, and when a multi-core method is used to accelerate this, only a minimum amount of resources should be added to deliver sufficient resources to other tasks. This study contributes by presenting a practical multi-threaded algorithm for accelerating sound rendering in a mobile device environment. For this purpose, three methods are included.

First, we used Guide mode (G mode) to find combinations of hit-triangles likely to generate valid paths by shooting multiple rays from the listener. This is the basis for creating multithreaded algorithms. Second, we parallelized Early Reflection mode (ER mode), which handles early reflection, and Late Reverberation mode (LR mode), which handles late reverberation.

Since this method uses only two threads and does not continuously maintain CPU utilization, the memory usage and CPU utilization increase rates were not large. Finally, we showed a thread synchronization scheme suitable for our algorithm. Through this, we solved the race condition problem that occurs during parallel processing.

We implemented this on a Galaxy 20+ smartphone using a Qualcomm Snapdragon 865 chipset equipped with the Adreno 650 GPU. We verified the performance by increasing the number of sound sources in various scenes. As a result, the performance of the proposed multithreaded method was about 1.77 times better on average than that of the single-threaded method. In addition, the increase rates (%) of the proposed method (memory usage, CPU utilization) were 1.07 and 0.87 on average, respectively, compared to the single-threaded method. This shows that our algorithm can be easily applied to the mobile device environment.

## 2. Related Work

This section gives an overview of sound propagation algorithms in the last few decades and their components.

### 2.1. Sound Propagation

Wave-based numerical methods calculate IRs by solving wave equations, which are usually second-order partial differential equations. Although such methods are accurate, they are very slow, so they are not suitable for interactive applications. Despite the fact that many studies have been conducted to accelerate the algorithm to solve this problem [11,12,13,14], considerable time and resources are still required, and the conditions remain limited.

The GA method has also been studied extensively. It has covered large scenes with many objects based on beam [15], frustum [16], or ray [17] tracing Among these, the ray tracing technique has recently been developed in both software and hardware. Therefore, most sound propagation algorithms supporting dynamic scenes are proposed based on ray tracing [18].

Various techniques to accelerate ray tracing-based GA algorithms have been proposed. The source clustering method, which combines sound sources under certain conditions to process many sound sources, improved the performance of the sound propagation algorithm by about 1.9 times based on 200 sound sources [8]. Backward ray tracing, which shoots rays from the listener rather than the sound source, lowered the cost of sound propagation sub-linearly [17]. A visibility graph to handle high-order reflection and diffraction was put forward [19]. An algorithm for quickly finding high-order diffraction paths using the A* pathfinding algorithm was put forth and showed to be about 568 times faster performance than the existing state-of-the-art method [20].

Acceleration methods using strong computing power have also been proposed, including a method of accelerating by assigning a thread to each sound source [6] by using a mixture of a CPU and a GPU [21] and by using a GPU [22].

However, the above methods utilize the powerful computing power of commodity CPUs or GPUs on a PC platform and as a result, maximize the corresponding computing resources. Hence, they utilize too many computing resources for sound rendering. In particular, sound rendering methods using GPUs are impractical because they take away resources for processing visual rendering. For these reasons, they are unsuitable for mobile devices with low computing power and low resources.

### 2.2. Sound Propagation Components

Sound propagation creates various sound effects through three components: direct sound, ER, and LR. Each component has different characteristics, which are the basis for creating various sound effects (see Figure 1).

Direct sound comes directly from the sound source to the listener and is the first of the components to arrive. Since it has the largest amplitude, it provides the maximum contribution to the distance and direction between the sound source and the listener.

ERs are the first echoes created after the arrival of the direct sound and are created through specular reflections and diffractions. LR is a very dense group of echoes and arrives last. It is created through high-order specular reflections or diffuse reflections.

ER and LR provide important perceptual cues about the space around the user, and many studies have been conducted to develop them. Specular reflection has been modeled using ray tracing [23], approximate volume tracing [24], and the image source method [25]. Among these, the image source method is the most accurate, so it is widely used in specular reflection modeling. We adopt the image source method for specular reflection.

There are two major methods for modeling diffraction: the Biot–Tolstoy–Medwin (BTM) [26] and the Uniform Theory of Diffraction (UTD) methods [27]. The BTM is more accurate than the UTD because it handles finite diffracting edges. However, it is not suitable for interactive applications because of the large amount of calculation. On the other hand, UTD is modeled assuming infinite diffracting edges. It is less accurate than BTM, but it is fast enough to be applied in interactive applications. For this reason, we adopt the UTD method for the diffraction.

Diffuse reflection is modeled using ray tracing [28], path tracing [29], and radiosity [30]. Since this generally requires a large amount of computation, it is not suitable for a mobile device environment.

## 3. Processing Flow and Analysis of Sound Rendering

This section introduces the sound rendering pipeline (Section 3.1) and the single-threaded sound propagation algorithm, which is the basis of the proposed algorithm (Section 3.2) and performance analysis (Section 3.3).

### 3.1. Sound Rendering Pipeline

Figure 2 shows the proposed sound rendering pipeline. It has two threads: a main thread that finds a valid path according to the location of the sound source and listener and calculates IRs and an auralization thread that creates the final sound using the IRs.

The main thread first imports scene data, such as geometry data and audio files, and then creates an acceleration structure (AS) such as a kd-tree or BVH for static objects through preprocessing. We adopt the AS as a kd-tree for fast search.

The auralization thread reads dry audio (PCM) as needed for every frame of the audio files imported by the main thread. Then, IRs received from sound propagation and the dry audio are convoluted to generate the final output signal, which is output to an output device (speakers or headphones). The above process is repeated through an auralization loop.

### 3.2. Single-Threaded Sound Propagation Algorithm

The proposed algorithm is implemented based on a single-threaded sound propagation algorithm. It is GA-based and uses ray tracing algorithms to create sound effects such as ER or LR. The ER is created through specular reflections (up to four-order) based on the image source method and edge diffractions (up to two-order) based on the UTD, and the LR is created through specular reflections (four-order).

Figure 3 shows a flowchart of the single thread sound propagation algorithm, images of the sound propagation modes included in the algorithm, and ray tracing processing in each mode.

The sound propagation algorithm is processed in the order of build acceleration structure, PathCache mode (PC mode), direct/transmission mode (DT mode), ER mode, and LR mode. Each step is as follows.

First, build acceleration structure build acceleration structure is a step of updating the kd-tree for dynamic objects. This enables the sound propagation algorithm to process dynamic scenes. Next, the sound propagation modes are performed. Those are the steps to create sound effects through ray tracing processing and include PC mode, DT mode, ER mode, and LR mode.

The PC mode is a step of finding valid reflection or diffraction paths in the current frame through propagation path caching. In other words, this process searches for valid paths in a path–cache–buffer where valid paths found in the previous frame are stored based on the location of the changed sound source and listener in the current frame.

Ray tracing algorithms create frame coherency issues due to the random directionality of the rays. To avoid such issues, propagation path caching is used in many interactive sound propagation algorithms [6,15].

DT, ER, and LR modes are steps for generating direct sound, ER, and LR, respectively. They find valid paths through the ray tracing processing and repeat the number of sound sources, the number of guide rays shot from listeners, and the number of source rays shot from sound sources, respectively.

All rays are processed through the ray tracing processing, in the order of ray generation, traversal and intersection (TnI), propagation path validation (PPV), and IR calculation. This is repeated for the maximum depth of the ray defined in the sound propagation. Each processing step is as follows.

Ray generation generates guide rays in PC and ER modes and source rays in DT and LR mode through random spherical sampling. TnI performs traversal to find hit-triangles using guide and source ray and then ray-triangle intersection tests. If the intersection tests are successful, PPV is conducted.

PPV finds valid paths through the validation test, as shown in Figure 4, based on the hit-triangles found by TnI. Then, the IR calculation describes the propagation effect by calculating the IRs between the sound source and the listener. It supports four frequency bands (0–250 Hz, 250–100 Hz, 1000–2000 Hz, 2000–4000 Hz) for each listener–source pair.

IRs of sound propagation modes have attenuation parameters of direction, delay, and frequency. The delay is calculated by dividing the length of the paths by the sound velocity, and the attenuation parameters are calculated by accumulating attenuation based on distance and frequency dependent wall absorption coefficients.

LRs’ IRs require additional parameters. We employ the widely used Eyring model [31] as the LR model. The parameters of this model are the volume of the room, the total absorbing surface area of the room, and the average absorption coefficient of the surfaces. They are computed using hit-triangles found by the guide and source rays. The IRs with the above information encoded are passed to the auralization thread to create the final sound.

### 3.3. Performance Analysis of Sound Propagation Modes

To effectively accelerate the sound propagation algorithm, it is essential to find which part of the existing single-threaded sound propagation algorithm is the bottleneck. To do so, we analyzed the performance of the sound propagation modes, which are the core of the sound propagation algorithm.

We performed the sound propagation algorithm in four scenes as shown in Figure 5 with a Galaxy 20+ smartphone using a Qualcomm Snapdragon 865 chipset equipped with the Adreno 650 GPU. In addition, we used eight static sound sources to increase the performance load, and shot 1024 guide and source rays, respectively.

Table 1 shows the performance of each sound propagation mode for eight sound sources. All the scenes spend a lot of time in ER and LR modes and relatively little time in PC and DT modes. Since more than 96% of the total time is spent in ER and LR modes, they clearly have many bottlenecks. For this reason, it is essential to accelerate the corresponding modes to improve the performance of the sound propagation algorithm, and we propose a multi-threaded sound propagation algorithm to overcome this problem.

## 4. Proposed Multi-Threaded Sound Propagation Algorithm

This section introduces the proposed multi-threaded-based techniques and structures to improve the performance of sound propagation algorithms. For this purpose, additional and modified sound propagation modes (Section 4.1) and synchronization methods (Section 4.2) are described.

### 4.1. Multi-Threaded Sound Propagation Algorithm

To apply GA-based sound rendering to interactive applications, it is very important to improve the performance of the sound propagation algorithm. However, since sound propagation algorithms are generally implemented based on ray tracing, it is very challenging to do so.

In particular, the cost of ER and LR increases rapidly with the number of valid paths and sound sources, which makes it much more difficult for them to perform at real-time rates. We propose a multi-threaded sound propagation algorithm to improve its performance.

Our basic idea is to accelerate the algorithm by performing multi-threaded ER and LR. For this, the single-threaded sound propagation algorithm is modified, and a new sound propagation mode is added to enable multi-threaded execution.

Figure 6 shows the flowchart of the proposed multi-threaded sound propagation algorithm. This is executed in the order of build acceleration structure, DT mode, and G mode, and then ER and LR modes are processed in parallel through two threads. Finally, the IRs from the two parallelized modes are merged through merge IRs. This is finally delivered to an auralization thread, and the algorithm is terminated.

The proposed algorithm has three newly proposed techniques for the parallelization of ER and LR. First, G mode, a key mode for parallelizing the sound propagation algorithm, is newly added. The goal of G mode is to find combinations of hit-triangles that are likely to be valid paths around the listener.

G mode has two stages: Step 01, which finds combinations of hit-triangles, and Step 02, which sorts the found combinations and removes duplicate elements (See Figure 7). The detailed process is as follows.

G mode shoots as many rays as the maximum number of guide rays set by the user to find combinations. The origin of the ray is set to the position of the listener, and the direction of the ray is calculated through spherical random sampling. Next, through ray tracing processing, G mode finds combinations of hit-triangles based on the ray. Then, the found combinations are stored in the combinations buffer.

Based on the found combinations, a sort is performed for each depth using a merge–sort based on an index of hit-triangles in the combinations. Then, duplicate combinations are removed by looping as many times as the number of combinations. The combinations made in this way are delivered to ER and LR modes. The pseudocode for G mode is summarized in Algorithm 1.
**Algorithm 1** G Mode
1: Rn←Ray, index:n={0,1,2…, N−1} , ***N*** = maximum number of guide ray 2: TB←Tree build data3: L←Listener4: CHT← Combinations of hit-triangles5: CB← Combinations buffer6: **procedure** Guide mode (L, TB, CB)7: **Step 01**: Finds combination of hit-triangles8: **for**
$R\in \left\{R_{0},\cdots R_{n-1}\right\}\;do\;$
9:  R← Set origin position (position of *L*) and random direction10:  CHT← Ray tracing processing (R, TB)11:  **if**
CHT is valid **then**12:   CB← Add CHT
13:  **end if**  14: **end for**15: **Step 02**: Sorts and removes duplicate combination of hit-triangles16: **for**
*d* = 0 to 3 **do** // depth loop17:  Merge-sort CHT that have depth *d* in CB18: **end for**19: **for**
*i* = *0* to *N − 1, j = 0 to N − 1*
**do** // *N* is number of combinations20:  **if**
CHTi is equal to CHTj
**then**21:   j←j+122:  **else**23:   Remove from CHTi+1 to CHTj−124:   i←j25:   j←j+126:  **end if**27: **end for**28: **end procedure**


Second, the ray tracing processing of ER and LR modes is changed, and PC mode is removed. In particular, ER mode typically finds valid paths while performing an amount of work in proportion to the maximum number of guide rays. However, the proposed method precalculates combinations of hit-triangles that are likely to be valid paths in G mode. For this reason, the ray tracing processing of the modes used in single-threaded algorithm is not suitable for our multi-threaded method, so it needs to be modified. The work processed in PC mode is processed by the newly added merge-hit-triangles in G mode and setup-hit-triangles in ER mode.

Figure 8 shows the flowchart of ER mode. It calculates additional information for calculating IR based on the combinations of hit-triangles generated by G mode and then generates IRs for ER. The steps of the processing are changed compared to the single-threaded method: a setup-hit-triangles step is added, and ray generation and TnI steps are removed because combinations of triangles are presearched in G mode.

The detailed processing process of setup-hit-triangles is as follows. It receives Cn (combinations of triangles) imported from G mode where 0 ≤ n ≤ N (number of combinations)—1, *L* (listener), and *S* (sound source) serve as input.

First, a merge–sort is performed on the combinations in Cn and the combinations in a path–cache–buffer of S. This is the same as Step 02 seen in G mode through which duplicate combinations are removed.

Then, additional information is calculated for T (triangles) in each combination while looping through Cn. To complete this, setup-hit-triangles determines the type for each T. The type variable indicates what kind of path will be created and includes reflection, diffraction, and none. If S is positioned toward the normal side of T, the type of S is reflection; otherwise, it is diffraction. If T is invalid, the type of T is determined to be none.

Through this, setup-hit-triangles determines what information needs to be additionally calculated for T. If T’s type is reflection, setup-hit-triangles calculates listener mirror positions for the image source method. Conversely, if it is diffraction, it computes edges information (edge point, edge vector) for UTD. The pseudocode for setup-hit-triangles is summarized in Algorithm 2.
**Algorithm 2** Setup-Hit-Triangles1: Cn← Combination of triangles by G mode, index:i={0,1, …, N−1}2: L  ← Listener3: S  ← Sound source4: PC← PathCacheBuffer of S5: Td← Triangle, depth:d = {0,1,2,3}6: TP(Td)← Type of Td= {Reflection, Diffraction, None}7: **procedure** Setup-hit-triangles (Cn, L, S)8**:**  Cn← Merge sort combinations in Cn and combinations in PC9:  **for** C∈{C0,⋯Cn−1} do10:   **for** T∈{T0,⋯T3} do11:    TP(Td)←GetTypeOfTriangle(Td, S) 12:    **if** TP(Td) is a reflection type **then**13:     Calculate image mirror positions based on Td and L14:     T← update the image mirror positions15:    **else if**
TP(Td) is a diffraction type **then**16:     Calculate edge information based on Td, S, and L17:     T← update edge information18:    **else** // None19:     **continue**20:    **end if**21:   **end for**22:  **end for**23: **end procedure**


Then, PPV in ray tracing processing finds a valid path among combinations as in Figure 4. After that, it is processed in the same way as the single-threaded algorithm. Through this, the IR for ER is created and passed to the merge-IRs step in Figure 6.

LR mode does not differ significantly from the existing single-threaded method, but the method of generating combinations for calculating IR is slightly different (see Figure 9). Since the single-threaded method proceeds sequentially, IRs are calculated immediately whenever source rays are shot one by one in LR mode after the combination for the listener is calculated in ER mode.

However, in the multi-threaded method, since ER and LR mode are divided into two threads, IRs cannot be calculated immediately in LR mode. Thus, when a valid path (combination of hit-triangles) is found by PPV, it temporarily stores the valid path without calculating the IR immediately.

In addition, when ray tracing processing is finished, combinations are merged through the merge-hit-triangle step as in G mode (Step 02) based on the combinations by LR mode and combinations imported from G mode. At this time, the same triangles included in both combinations are designated as triangles that can be contributed to the Eyring model, and IRs are calculated based on the triangles.

The final change is to separate the IRs memory for ER and LR mode. Multi-threaded algorithms cause data races due to shared resources. To prevent this, a synchronization lock such as a mutex is required, but the cost of such a lock degrades the performance of the algorithm.

To reduce this cost, we remove locks for the synchronization in the IR buffer that stores IRs in each thread, and separate buffers for ER and LR to store IRs. If two threads create IRs and store them in respective IR buffers, the IRs in the two buffers are merged through the merge-IR step, as shown in Figure 6.

### 4.2. Thread Synchronization

As the proposed algorithm performs parallel processing through two threads, thread synchronization is essential. We use three functions (Wait, SetEvent, ResetEvent) for thread synchronization. Wait (Object) is a function that waits until a specific event object becomes true. Set/ResetEvent (Object) are functions that change the signal of an event object to true/false. For example, if there is an event object called T0 and Wait (T0) is called, the thread waits until SetEvent (T0) is called. Conversely, ResetEvent (T0) blocks the corresponding thread.

We perform thread synchronization as shown in Figure 10. Thread01 performs DT, G and ER modes and thread02 performs LR mode. We divide LR mode into ray tracing processing for LR and (merge-hit-triangles + IR calculation) to increase the parallelism of the algorithm.

Thread02 starts after SetEvent (LR0) is called on thread01. Then, thread02 waits until G mode finishes. If SetEvent (LR1) is called in thread01, merge-hit-triangles and IR calculation are performed in thread02. In thread01, merge-IRs are executed when IR calculation of LR is finished.

## 5. Experimental Results

This section introduces the experimental environment and settings (Section 5.1) and describes the experiments performed to determine the appropriate number of rays for load-balancing of the proposed algorithm (Section 5.2). In addition, it evaluates the performance of the proposed multi-threaded algorithm through a performance comparison with the single-threaded algorithm (Section 5.3) and assesses algorithm overhead by determining the memory usage and CPU utilization of single-threaded and multi-threaded algorithms (Section 5.4).

### 5.1. Experimental Setup

We implemented the sound propagation algorithm in the form of a native plug-in (.so, .dll) and connected it to the Unity game engine to conduct experiments (see Figure 11). The performance of the sound propagation algorithm varies greatly depending on the ray depths and the number of triangles and valid paths, which are inherently changed by the characteristics of the scenes.

For this reason, as shown in Figure 5, we adopted two indoor scenes and two hybrid scenes mixed with indoor and outdoor. Sibenik, concerthall, and angrybot scenes are static scenes, and racelake is a dynamic scene. We conducted experiments with the sound source and listener stopped to give a performance load to the sound propagation algorithm, and the experiment device was a Galaxy S20+.

### 5.2. Load-Balancing

When the sound propagation algorithm-based ray tracing shoots more rays, it finds more valid paths, making it more likely to generate appropriate audio that matches the visual rendering. However, shooting a large number of rays (10k, 100k) degrades sound rendering performance.

In addition, in a multi-threaded algorithm, appropriate load-balancing between threads performing tasks is essential. We needed to appropriately adjust the number of rays that most affect the performance of the two threads to find the optimal load-balancing in our algorithm. Thus, we conducted an experiment to find an appropriate ratio between the number of guide rays used in G and ER modes and the number of source rays used in LR mode.

We set the number of sound sources, the number of guide rays, and the maximum depth to 8, 1024, and 4, respectively, in the Sibenik scene, which is the worst case of the scenes. After that, we measured the increase rate of the performance of the multi-threaded algorithm compared to that of the single-threaded algorithm by increasing the number of source rays (64 to 4096) for each sound source (see Figure 12). The load-balancing of the two threads improved with the performance increase.

The experimental results showed that the performance increase rate gradually rose when the number of source rays went from 64 to 1024, and the performance increase rate was the highest when the number of source rays was 1024.

This means that when the number of source rays is less than 1024, LR mode must wait for a certain time until ER mode is finished because the throughput of LR mode is greater than that of ER mode. This waiting time causes performance degradation.

Conversely, when the number of source rays is 1024 to 4096, the throughput of LR mode is higher than that of ER mode. Because of this, ER mode must wait for a certain time until LR mode is finished, so the rate of increase in performance gradually decreases. That is, the proposed algorithm shows the best performance and the best load-balancing when the ratio of the number of guide rays to the total number of source rays is about 1:8 in the worst case.

### 5.3. Performance

Table 2 shows the performance comparison of the single-threaded and multi-threaded algorithms for the four scenes. We set the maximum depth, the number of guide rays, and the number of source rays to 4, 1024, and 1024, respectively. We measured the number of valid reflection and diffraction paths and the average frame time for 100 frames while increasing the number of sound sources (1, 2, 4, and 8) for each scene.

The experimental results showed that the performance increase rate for each scene based on 8 sound sources was 84.96% in sibenik, 104.67% in concerthall, 54.95% in angrybot, and 64.46% in racelake. These showed that the performance of the proposed multi-threaded method was on average 77.26% better than the of the single-threaded method.

### 5.4. Memory Usage and CPU Utilization

We measured CPU utilization and memory usage to assess the overhead introduced by our multithreaded algorithm and compared it to that of the single-threaded algorithm. We used Snapdragon Profiler as the program for measurement.

In the case of the CPU utilization experiment, we fixed the FPS of the two comparison groups for a fair experiment and set the maximum depth, number of sound sources, number of source rays, and number of guide rays to 4, 8, 1024, and 1024, respectively. We then measured the average CPU utilization for 30 s.

Table 3 shows the average CPU utilization and the difference between the single-threaded and multi-threaded algorithms. The experimental results showed that the CPU utilization of the single-threaded algorithm was lower than that of the multi-threaded algorithm in all scenes.

However, the difference (%) between the two utilizations was 1.00, 1.40, 0.38, and 0.71 in sibenik, concerthall, angrybot, and racelake, respectively. In other words, our algorithm does not use many CPU resources even though it uses a multi-threaded method. This is because it uses only two threads and does not constantly use CPU resources.

In the case of the memory usage experiment, we used the same conditions as in the CPU utilization experiment and measured only the memory usage used in the sound propagation algorithms for 30 s while increasing the number of sound sources (1–8) in sibenik. 

Table 4 shows the memory usage and the difference between the two algorithms. The experimental results showed that the difference in memory usage (MB) was 1.80, 1.49, 0.63, and 0.39, respectively, depending on the number of sound sources (1–8). This can be said to have low memory overhead because multi-threaded algorithms do not increase memory usage significantly.

As can be seen from the above experimental results, the proposed multi-threaded algorithm not only has higher performance than the single-threaded method, but also is more suitable for the mobile device environment as it minimizes the increase in memory usage and CPU utilization.

## 6. Conclusions

This paper proposed a multi-threaded sound propagation algorithm to improve the performance of sound propagation algorithms in mobile devices. To achieve this, we mainly used three methods. First, we performed what is called G mode for parallel task processing. This enabled ER and LR modes to perform in parallel by finding combinations of hit-triangles likely to create valid paths by shooting multiple rays from the listener.

Second, we split the processing into two threads: ER mode, which produces early reflection, and LR mode, which produces late reverberation. Finally, we solved the problem of the race condition by applying a suitable thread synchronization technique.

Based on this, we showed that the two modes can be simultaneously processed in parallel to improve the performance of the sound propagation algorithm. In addition, since this method uses only two threads and does not increase the memory usage or CPU utilization rate compared to the single-threaded method, we found that it is suitable for application in the mobile device environment.

We verified the performance, memory usage, and CPU utilization of the proposed algorithm in various scenes. The experimental results showed that the performance of the multi-threaded method was about 1.77 times better than that of the single-threaded method. Moreover, the average increase rates (%) in terms of memory usage and CPU utilization of the multi-threaded algorithm were 1.07 and 0.87. These increase rates were with negligible overhead, indicating no burden of additional overhead. That is, it showed that our algorithm is suitable for application in a mobile device environment and exhibits a certain increase in performance.

## Figures and Tables

**Figure 1 sensors-23-00973-f001:**
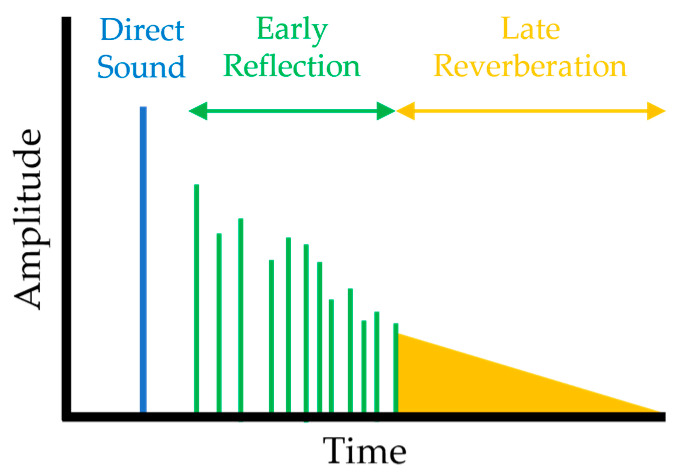
Amplitude of each component over time (direct sound, ER, LR).

**Figure 2 sensors-23-00973-f002:**
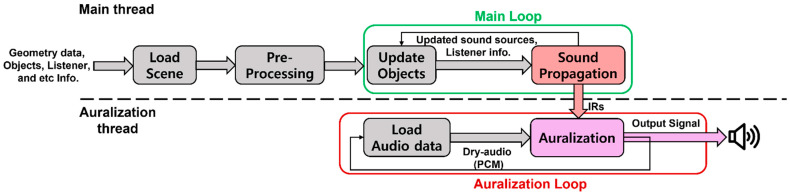
Sound rendering pipeline: main thread that handles sound propagation and auralization thread that creates final audio output signal.

**Figure 3 sensors-23-00973-f003:**
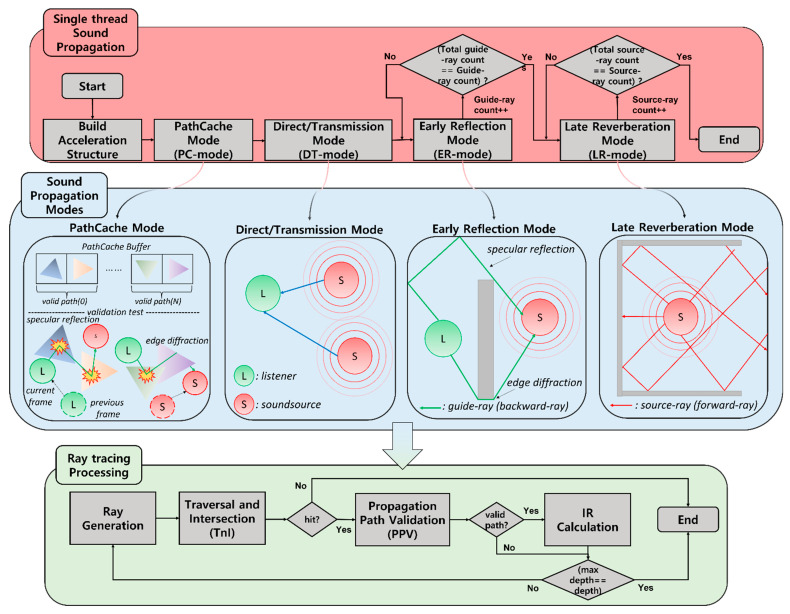
Flowchart of single-threaded sound propagation algorithm (**top**), descriptions of sound propagation modes (**middle**), and ray tracing processing (**bottom**).

**Figure 4 sensors-23-00973-f004:**
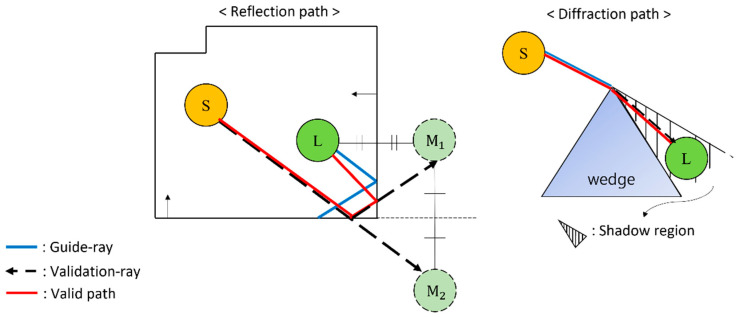
Two examples showing validation test for 2-order specular reflection path (**left**) and 1-order diffraction path (**right**), respectively. Validation test for reflection path: The listener (L) is reflected recursively over the sequence of reflecting hit-triangles. Through this, listener mirror images (M1, M2) are created. Then, occlusion tests are performed while a validation ray shooting from the sound source (S) to the last listener image (M2) is specularly reflected back to the L. If the tests are successful, valid reflection paths are created. The validation test for the diffraction path: Check if L is in the shadow region based on the wedge containing the hit-triangle. If L is within the region, the three edge points closest to the LS→ straight line are calculated for the three edges included in the hit-triangle. Then, occlusion tests are performed through the validation rays shooting from the edge points to L. If the tests are successful, valid diffraction paths are created.

**Figure 5 sensors-23-00973-f005:**
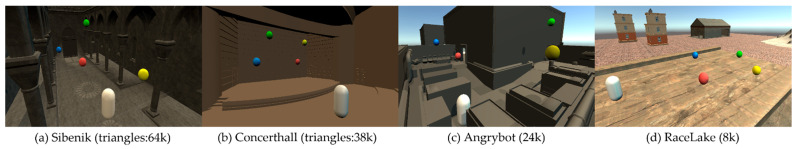
Four benchmark scenes: (**a**) sibenik (indoor)—64k; (**b**) concerthall (indoor)—38k; (**c**) angrybot (indoor + outdoor)—24k; (**d**) racelake (indoor + outdoor)—8k.

**Figure 6 sensors-23-00973-f006:**
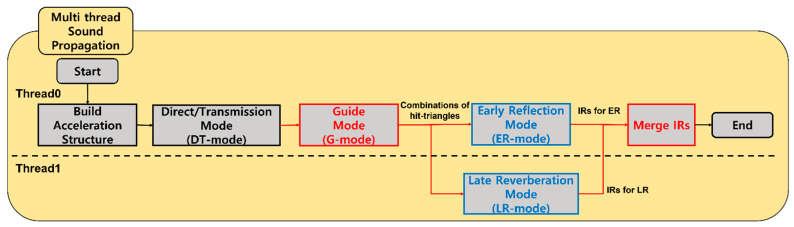
Proposed multi-threaded sound propagation algorithm: new stages (red boxes), modified stages (blue boxes).

**Figure 7 sensors-23-00973-f007:**
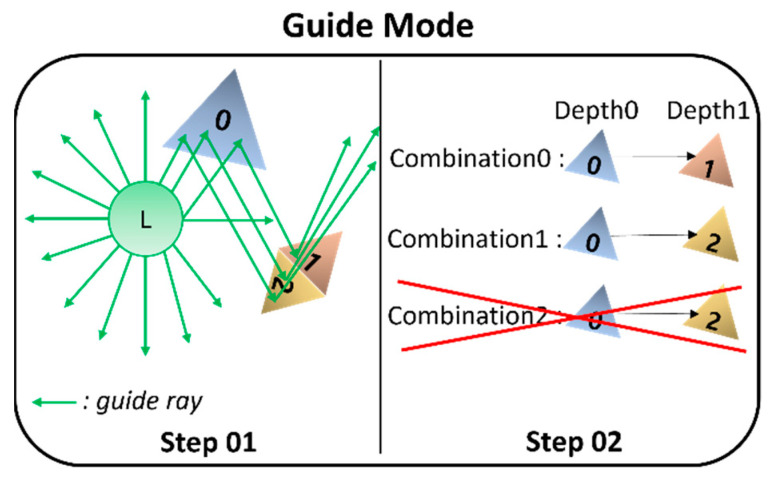
Description of G mode: Find combinations of hit-triangles through guide ray (Step 01) and remove duplicate combinations among combinations using a merge sort (Step 02).

**Figure 8 sensors-23-00973-f008:**
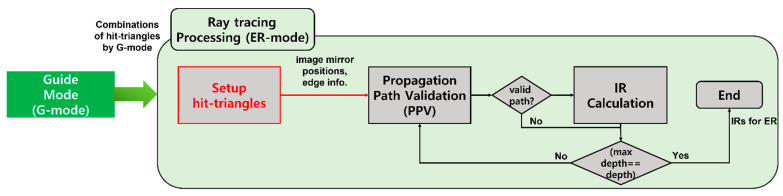
Flowchart of ER mode for one sound source modified to suit the multi-threaded method: new step (red box).

**Figure 9 sensors-23-00973-f009:**
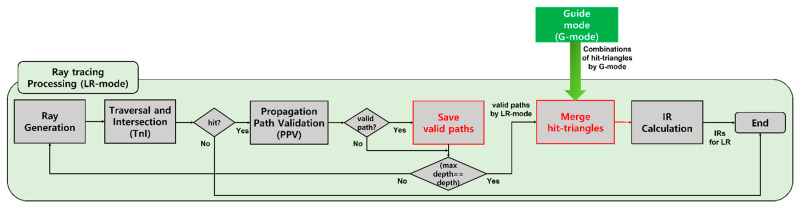
Flowchart of LR mode for one sound source modified to suit the multi-threaded method: new steps (red boxes).

**Figure 10 sensors-23-00973-f010:**
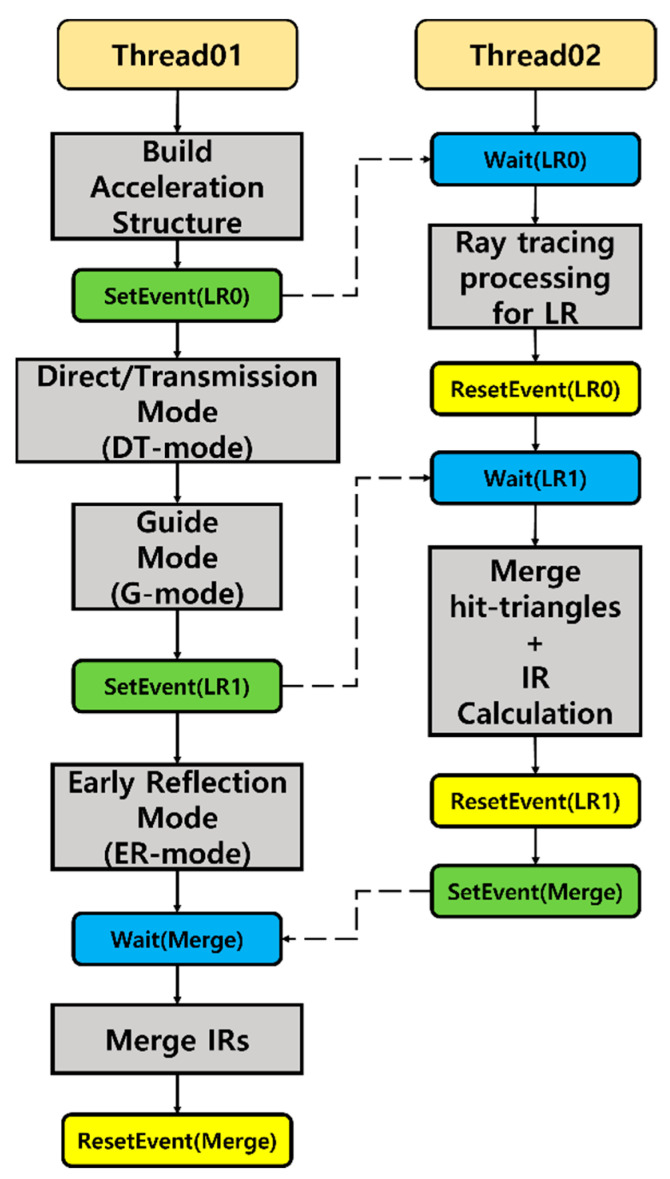
Thread synchronization: Thread01 and thread02 perform synchronization through SetEvent, ResetEvent, and Wait functions.

**Figure 11 sensors-23-00973-f011:**
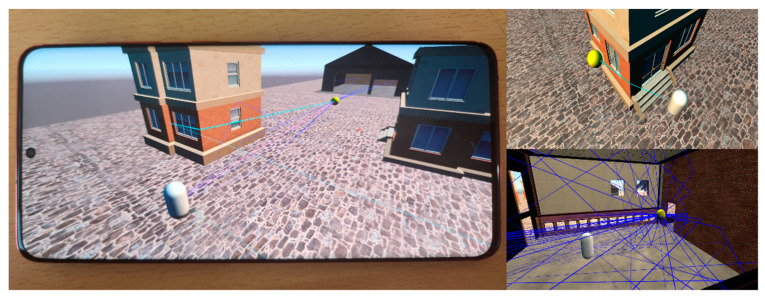
Sound propagation algorithm running on Galaxy S20+.

**Figure 12 sensors-23-00973-f012:**
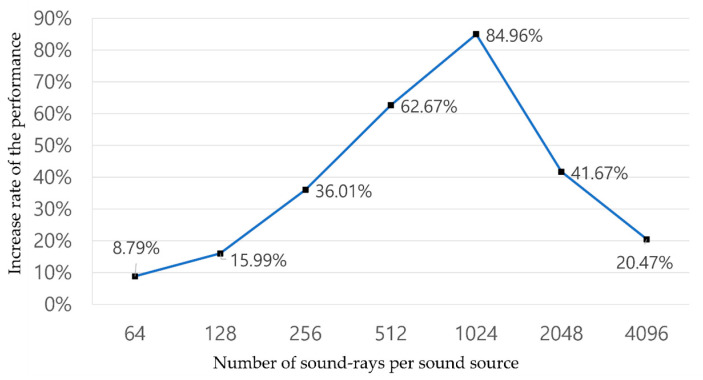
Increase rate of multi-threaded performance compared to single-threaded performance.

**Table 1 sensors-23-00973-t001:** Performance by modes of single-threaded sound propagation algorithm for eight sound sources (frame time: ms).

	PC Mode	DT Mode	ER Mode	LR Mode	Total Time
Sibenik	8.15	0.00	125.19	168.90	302.6
Concerthall	10.82	0.00	142.86	161.91	315.6
Angrybot	2.06	0.00	38.59	60.45	101.1
RaceLake	8.04	0.00	78.91	131.44	218.4

**Table 2 sensors-23-00973-t002:** Performance comparison of single-threaded and multi-threaded algorithms for four scenes.

	SoundSource	Reflection Path (max: 4-Order)	Diffraction Path (max: 2-Order)	Single-Threaded Frame Time (ms)	Multi-Threaded Frame Time (ms)	Increase Rate (%)
Sibenik	1	36	7	57.2	32.4	76.54
	2	77	18	90.2	49.2	83.33
	4	153	20	162.6	82.8	96.38
	8	315	53	302.6	163.6	84.96
Concert hall	1	62	0	57.8	35.8	61.45
	2	128	2	95.0	50.4	88.49
	4	266	8	167.4	84..8	97..41
	8	562	16	315.6	154.2	104.67
Angrybot	1	11	2	25.4	14.4	76.39
	2	30	9	37.8	19.8	90.91
	4	53	15	62.2	34.6	79.77
	8	84	28	100.1	64.6	54.95
RaceLake	1	75	0	45.8	25.2	81.75
	2	180	0	78.6	50.6	55.34
	4	295	2	139.8	97.4	43.53
	8	608	5	218.4	132.8	64.46

**Table 3 sensors-23-00973-t003:** Average CPU utilization comparison of single-threaded and multi-threaded algorithms.

Average CPU Utilization (%)			
Scene	Single-Threaded	Multi-Threaded	Difference
Sibenik	16.30	17.30	1.00
Concert hall	17.80	19.20	1.40
Angrybot	11.20	11.60	0.38
RaceLake	14.60	15.40	0.71

**Table 4 sensors-23-00973-t004:** Average memory usage comparison of single-threaded and multi-threaded algorithms in sibenik.

Average Memory Usage (MB)			
Sound Source	Single-Threaded	Multi-Threaded	Difference
1	378.63	380.43	1.80
2	382.04	383.52	1.49
4	386.16	386.79	0.63
8	395.49	395.88	0.39

## Data Availability

Not applicable.

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
