# Peer review of "Multi-Threaded Sound Propagation Algorithm to Improve Performance on Mobile Devices"

_sensors, 2023, doi:10.3390/s23020973_

Round 1

Reviewer 1 Report

1.       The abstract is long and NOT satisfactory. It should contain the following parts:

  i. The importance of or motivation for the research.

  ii. The issue/argument of the research.

  iii. The methodology.

  iv. The result/findings.

  v. The implications of the result/findings.

2.       Explain the motivation pin points to make them understandable for readers. It is bit confusing. Rewrite the motivation point if possible.

3.       It is recommend to add a detailed system process flow of the proposed methodology.

4.        Verify all equations if they are added correctly.

5.       Figure of proposed model need refinement try to sketch neatly.

6.       Authors should add an algorithm to explain the proposed work and explain it in the proposed work section extensively.

7.       Is the proposed system secure enough and sustainable to apply in distributed environment? If yes, kindly approach with the below work and preferably include in the related work

a.       Bdtwin: an integrated framework for enhancing security and privacy in cybertwin-driven automotive industrial Internet of things. IEEE Internet of Things Journal.

b.      Permissioned blockchain and deep-learning for secure and efficient data sharing in industrial healthcare systems. IEEE Transactions on Industrial Informatics.

c.       Large-scale data storage scheme in blockchain ledger using ipfs and nosql. In Large-Scale Data Streaming, Processing, and Blockchain Security (pp. 91-116). IGI Global.

Reviewer 2 Report

The authors proposed a Multi Thread Sound Propagation Algorithm to Improve Per- 2 formance on Mobile Devices. The problems discussed in the paper are interesting and well-motivated. The authors clearly describe their system model and design goals, and the paper is well presented and relatively complete. However, the authors need some revisions as follows:
1. Section 1 Introduction: In the introduction, the aim of the paper is clearly written. However, the author should emphasize and add more of the issues and the significance of this research.
2. Section 2 Related Work: The authors should discuss and cite more recent related papers, such as those published in 2021 and 2022. Also, include one relative comparosn table of the proposed study with the state-of-the-art approaches. 
3. Writing: More thorough proofreading of this work would be beneficial. I found many typos. Please check them very carefully.
4. Section 4 Proposed Protocol: Section 4 proposes a new approach. The authors should check all equations/notations/definitions again. An overview of the proposed protocol may be provided so readers can easily follow the scheme.
5. Section 5 Result and Discussion: The comparison table 
of single-threaded and multi-threaded algorithms for four scenes (Table 2) shows that s the performance comparison of single-threaded and multi-threaded algorithms for four scenes does not included overheads. However, the authors did not discuss it in detail. The advantages of the proposed study over other should be explained briefly

6. Recheck all of your references, there are some mistakes.
7. Some of your references are not eligible for using in scientific paper.
8. Comparisons are not satisfactory and comprehensive, besides state of art works are not completely covered in them.
9. What is Take Away section! What's for? if you like to make a section for future works, write it instead of this messy way of organization.
10. Conclusion section is not comprehensive in any aspect which it should be

11. Why there is increase in the rate of the performance of the multi threaded algorithm compared to that of the single-threaded algorithm by increasing the number of source-rays (64 to 4096) for each sound source. Not clear from the explanation. 

Reviewer 3 Report

The authors missed only one thing in the article 

1. Please list the contributions of the proposed article in the introduction sections. 

Round 2

Reviewer 1 Report

The authors have incorporated all the changes.

Reviewer 2 Report

The authors have revised the paper as per the comments and no further comments 

Great work by the authors

Best